# Application of Intelligent Control in Chromatography Separation Process

**Chao-Fan Xie [1], Hong Zhang [2]** and **Rey-Chue Hwang [3],***

1   School of Big Data and Artificial Intelligence, Fujian Polytechnic Normal University, Fuqing 350300, China; chaofanxie@hotmail.com
2   Key Laboratory of Nondestructive Testing, Fujian Polytechnic Normal University, Fuqing 350300, China; zhhgw@hotmail.com
3   Department of Electrical Engineering, I-Shou University, Kaohsiung 84001, Taiwan
*   Correspondence: rchwang@isu.edu.tw; Tel.: +886-0928722593

**Abstract:** Chromatographic separation plays a pivotal role in the manufacturing of chemical products and biopharmaceuticals. This technique exploits differences in distribution between stationary and mobile phases to separate mixtures, impacting final product quality. Simulated moving bed (SMB) technology, recognized for its continuous feed, enhances efficiency and increases production capacity while reducing solvent and water consumption. Despite its complexity in controlling variables like flow rates and valve switching times, traditional control theories fall short. This study introduces an intelligent fuzzy controller resembling an approximate neural network (NN) for SMB control. Simulation results demonstrate the controller's effectiveness in achieving desirable outcomes for the SMB system.

**Keywords:** chromatographic separation; SMB; fuzzy; neural network





## 1. Introduction

The chromatographic separation principle relies on the equilibrium distribution of substances between stationary and mobile phases. Different substances move at different rates with the mobile phase, causing mutual separation in the stationary phase.

SMB is a liquid chromatographic separation device based on adsorption. The mixture dissolves in a solvent, is injected into a high-pressure column, and components move through the stationary phase based on their distribution. Weakly interacting components quickly exit, while strongly interacting ones take longer. Chromatography is widely used in chemical and biomedical industries, especially in biomedical column chromatography. SMB technology, enabling continuous separation, is valued for increased productivity, and reduced solvent use, making it a clean technology in biopharmaceutical manufacturing. Multi-column SMB is seen as essential in future biochemical pharmaceutical separation and purification processes [1–3].

SMB operation closely mirrors the traditional true moving bed (TMB) concept from chemical engineering (see Figure 1). In a column filled with solid adsorbent, a feed containing components A and B undergoes upward flow. The solid absorbs the adsorbate, followed by washing with a down-flowing mobile phase. This results in the upward movement of component A (stronger affinity) and the downward movement of component B (weaker retention).

Connecting multiple adsorption beds in series and configuring inlet and outlet points between them, including feed and sorbent inlets and three outlets (extract, raffinate, and sorbent for recycling), involves switching feed and withdrawal points clockwise to the next position after a set period. This simultaneous movement creates a counterclockwise flow of the solid. Continuously changing the positions of inlet and outlet points generates a counterclockwise flow of solid adsorbent, while the mobile phase maintains a clockwise

flow. This achieves continuous counter-current contact between the solid and the mobile phase, akin to the TMB process.

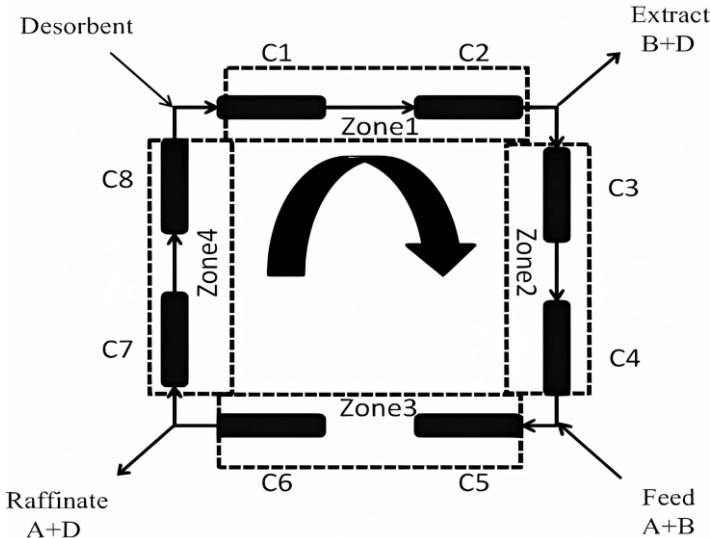

**Figure 1.** SMB operation process.

While the simulated moving bed (SMB) has enhanced chromatographic separation efficiency, its industrial implementation involves numerous parameters. Determining optimal parameters through trial-and-error experiments incurs significant costs. Researchers seek simulation-based quantitative analyses by examining mathematical models, with the general rate model and balanced diffusion model being primary choices [4]. The former, comprehensive but complex, closely mimics the actual process, while the latter, a simplified mechanism, is effective at lower component concentrations. Further model subdivisions are possible [5,6].

In the realm of nonlinear control systems, traditional design relies on the Lyapunov stability theorem. However, for the highly nonlinear, sensitive, and discretely switched SMB system, finding an appropriate Lyapunov function is challenging. Designing a traditional nonlinear adaptive controller is formidable. Consequently, developing an effective control method for multi-column SMB systems remains a significant goal in biomedical and chemical applications.

Fuzzy control, widely applied in nonlinear control, proves effective for unknown or partially known systems. Neural networks' learning capability excels in identifying unknown systems. This study aims to design an approximate neural network-like fuzzy controller for SMB control, capable of learning system characteristics and modulating parameters effectively. This controller is expected to significantly contribute to practical SMB chromatographic separation.

For research into the control of complex nonlinear systems, with a focus on model-based studies rooted in both theoretical laboratory and practical applications, several control strategies are prominently involved. These include model predictive control (MPC), multi-level, multi-stage feedback control, and self-tuning control strategies. For instance, Klatt et al. introduced a model-based optimization control scheme for SMB chromatographic separations and its application in the separation of fructose and glucose [7]. Andrade Neto et al. presented a self-adjusting nonlinear MPC method for the enantiomeric separation of Praziquantel in an analog moving bed apparatus [8]. This control strategy aims to achieve an efficient and accurate separation of Praziquantel enantiomers while maintaining the desired purity levels through real-time adjustments of control parameters. In both studies, the controllers successfully maintained the purity levels at the desired setpoints, achieving 99% purity in the extract and 98.6% purity in the residual solution.

Nogueira et al. have proposed a nominally stable MPC controller, also known as infinite horizon model predictive control (IHMPC), for the control of the simulated

moving bed process applied to the separation of binaphthol enantiomers. This work lays the foundation for further advancements in the field of loop process dynamics and control [9]. Ju-Wen Lee introduced a simplified linear isothermal process model to estimate the process state of SMB chromatography. Within the moderately nonlinear range of the Langmuir isotherm, optimal setpoint conditions have been determined through a "one-switch-at-a-time" switching operation [10]. Yang et al. presented an optimization strategy based on an improved moving asymptote algorithm. [11]. Carols et al. proposed a method that combines wave theory and multi-model predictive control for the analysis and simulation of the dynamic characteristics of moving beds [12]. Suvarov et al. applied a self-regulating control strategy to adjust the purity and productivity of the raffinate, as well as the extraction flow rate, by modifying the spatial position of adsorption and desorption waves. This self-tuning control technology has been widely used in program-based control [13]. Maruyama et al. developed a multi-stage continuous control method for bypass-simulated moving beds (BP-SMBs) and demonstrated its feasibility through simulation results. The method effectively fits the set target yield value to the actual yield curve of the target substance and impurities [14].

In terms of simulating mobile bed optimization, Leibnitz et al. proposed a model-based optimization approach for the selection of an adsorbent in mobile bed processes. The method utilizes correlations between the structural properties of the adsorbent and the model parameters of a transport dispersive model [15]. Schulze et al. introduced the transient nonlinear wave propagation model (TWPM) for the dynamic modeling of multi-component distillation columns with variable holdup. They demonstrated the applicability of this model for optimization and control purposes in single-section distillation columns and simple air separation units [16]. In terms of deep learning applications, Woo-Sung Lee and Chang-Ha Lee utilized limited experimental parameters and real industrial data to develop mathematical dynamic models and data-driven machine learning approaches for evaluating SMB performance [17]. Marrocos et al. proposed a deep artificial intelligence structure with a nonlinear output error (NOE) structure and a nonlinear autoregressive with exogenous input (NARX) predictor. This structure serves as an online soft sensor to provide information on the main properties of an SMB chromatographic unit [18]. Hoon et al. applied a data-based deep Q network, which is a model-free reinforcement learning method, to train near-optimal control strategies for the SMB process [19]. This approach utilizes deep Q-networks and parallel numerical simulations to generate sufficient data for training control strategies for complex dynamic systems. For more details and further related studies, refer to the literature [5,20–26]. Typically, these studies are specific to particular equipment and separation materials rather than general solutions.

## 2. SMB Mathematical Discrete Model

Generally, traditional nonlinear model analysis primarily targets affine systems; however, the nonlinear nature of SMB exhibits three distinctive characteristics. (1) SMB control involves switching time parameters with discrete events, making it inherently more complex and nonlinear. (2) The control objective is not merely minimizing control output errors but rather ensuring that the component concentrations of the desired separated materials reach specific target ratios. In practical SMB operation, control parameters are often empirically determined. (3) SMB control variables exhibit strong coupling, and once control variables fall into an infeasible separation area, the system's output will appear uncontrollable.

In this study, firstly, the Crank–Nicolson method is employed for numerical computation of the SMB system to achieve more effective control. Secondly, the stability and error convergence of the discrete model are analyzed. Thirdly, sensitivity analysis of the regional velocity to pure monotonic intervals is conducted.

### 2.1. TMB and SMB Equation Model

The traditional SMB dynamic mathematical model is derived by referencing the TMB mathematical model. The definition of the parameters is presented in Table 1.

**Table 1.** Parameters of SMB system.

| Parameter | Nomenclature | Parameter | Nomenclature |
|:---:|:---:|:---:|:---:|
| $x(\text{cm})$ | Axial distance | $Q(\text{cm}^3\text{min}^{-1})$ | Volume flow rate |
| $k(\text{gL}^{-1})$ | Comprehensive mass transfer constant | $t(\text{second})$ | Time |
| $v^*(\text{cm min}^{-1})$ | Effect velocity of body | $D(\text{cm}^2\text{min}^{-1})$ | Effective dispersion coefficient |
| $u_s(\text{cm min}^{-1})$ | Solid flow rate | $\varepsilon$ | Bulk void fraction |
| $C(\text{gL}^{-1})$ | Mobile phase concentration | $i$ | Material index: A or B |
| $q(\text{gL}^{-1})$ | Solid phase concentration | $j$ | Column number: 1, 2, 3, 4, 5, 6, 7, 8 |
| $q^*(\text{gL}^{-1})$ | Solid phase concentration at equilibrium between solid phase and mobile phase | | |

The following is an overview of the SMB dynamic model mentioned in the literature [17–19]. For the TMB, the mass balance of the bulk phase is given by:

$$\frac{\partial C_{ij}}{\partial t} = D_i \frac{\partial^2 C_{ij}}{\partial x^2} - v_j \frac{\partial C_{ij}}{\partial x} - \frac{1-\varepsilon}{\varepsilon} k_i (q_{ij}^* - q_{ij}) \tag{1}$$

$$\frac{\partial q_{ij}}{\partial t} = \frac{\partial}{\partial x} u_s q_{ij} + k_i (q_{ij}^* - q_{ij}) \tag{2}$$

Therefore, the TMB and SMB can be converted to each other using the following conversion:

$$\frac{\partial C_{ij}}{\partial t} = D_i \frac{\partial^2 C_{ij}}{\partial x^2} - v_j^* \frac{\partial C_{ij}}{\partial x} - \frac{1-\varepsilon}{\varepsilon} k_i (q_{ij}^* - q_{ij}) \tag{3}$$

$$\frac{\partial q_{ij}}{\partial t} = k_i (q_{ij}^* - q_{ij}) \tag{4}$$

where $v_j^*$ is the velocity of SMB. Substituting Equation (4) into Equation (3), we obtain the following equation:

$$\frac{\partial C_{ij}}{\partial t} = D_i \frac{\partial^2 C_{ij}}{\partial x^2} - v_j^* \frac{\partial C_{ij}}{\partial x} - \frac{1-\varepsilon}{\varepsilon} \frac{\partial q_{ij}}{\partial t} \tag{5}$$

The adsorption equilibrium is represented by linear isotherms:

$$q_{ij} = H_i C_{ij} \tag{6}$$

The flow volume in each region must satisfy the following conditions:

$$\begin{aligned} &Q_I > Q_{IV}, Q_I > Q_{II}, Q_{III} > Q_{IV}, Q_{III} > Q_{IV} \\ &Q_I - Q_{IV} = Q_d, Q_I - Q_{II} = Q_x, Q_{III} - Q_{IV} = Q_r, Q_{III} - Q_{II} = Q_f \\ &v_j = \frac{Q_j}{\pi r^2} \end{aligned} \tag{7}$$

where $Q_d$, $Q_x$, $Q_r$, and $Q_f$ are, respectively, the solvent, extract, raffinate, and feed stream flow rates; they can also be selected as controllable variables. r represents the radius of column.

### 2.2. SMB Digitization via Crank–Nicolson Method

To achieve effective control of SMB, we conducted a numerical simulation analysis of the analogous substance diffusion process in the SMB system to perform a rigorous examination and verification:

Set $t_k = t_0 + ks$, $x_l = x_0 + lh$, $F = \frac{1-\varepsilon}{\varepsilon}$

$$\frac{\partial^2 C_{ij}}{\partial x^2} = \frac{\frac{\partial^2 C_{ij}}{\partial x^2}|_{t_k} + \frac{\partial^2 C_{ij}}{\partial x^2}|_{t_{k+1}}}{2}$$
$$= \frac{C_{ij}(x_{l-1}, t_{k+1}) - 2C_{ij}(x_l, t_{k+1}) + C_{ij}(x_{l+1}, t_{k+1}) + C_{ij}(x_{l-1}, t_k) - 2C_{ij}(x_l, t_k) + C_{ij}(x_{l+1}, t_k)}{2h^2} \tag{8}$$

$$\frac{\partial C_{ij}}{\partial x} = \frac{\frac{\partial C_{ij}}{\partial x}|_{t_k} + \frac{\partial C_{ij}}{\partial x}|_{t_{k+1}}}{2}$$
$$= \frac{C_{ij}(x_{l+1}, t_{k+1}) - C_{ij}(x_{l-1}, t_{k+1}) + C_{ij}(x_{l+1}, t_k) - C_{ij}(x_{l-1}, t_k)}{4h} \tag{9}$$

$$\frac{\partial C_{ij}}{\partial t} = \frac{C_{ij}(x_l, t_{k+1}) - C_{ij}(x_l, t_k)}{s} \tag{10}$$

$$\frac{\partial q_{ij}}{\partial t}(x_l, t_k) = H_i \frac{\partial C_{ij}}{\partial t}(x_l, t_k) \tag{11}$$

$$i = 1, \cdots M, j = 1, \cdots N$$

where $M$ and $N$ represent, respectively, the number of material and zones; here, $M = 2$, $N = 8$.

Substituting into the SMB system, $C_{ij}(x_l, t_k)$ denotes as $C_{ij}(l, k)$, and then, we can obtain:

$$(1 + FH_i + \frac{Ds}{h^2})C_{ij}(l, k+1) - (\frac{vs}{4h} + \frac{Ds}{2h^2})C_{ij}(l-1, k+1) + (\frac{vs}{4h} - \frac{Ds}{2h^2})C_{ij}(l-1, k+1)$$
$$= (\frac{Ds}{2h^2} + \frac{vs}{4h})C_{ij}(l-1, k) + (1 + FH_i - \frac{Ds}{h^2})C_{ij}(l, k) + (\frac{Ds}{2h^2} - \frac{vs}{4h})C_{ij}(l-1, k) \tag{12}$$

With the boundary conditions:

$$C_{ij}(x, 0) = C_{0ij} \tag{13}$$

$$\frac{\partial C_{ij}(x, t)}{\partial x}|_{x=l_s} = 0 \tag{14}$$

$$D_i \frac{\partial C_{ij}(x, t)}{\partial x}|_{x=0} = v_j[C_{ij}(0, t) - \overline{C}_{ij}^{\sec t}(t)] \tag{15}$$

where $C_{0ij}$ represents the initial concentration distribution inside the columns at $t = 0$. In Formulas (14) and (15), $l_s$ and 0 represent, respectively, the end and initial position of the column. In an SMB arrangement, the column inlet concentration $\overline{C}_{ij}^{\sec t}(t)$ depends on the section and the location of the column within the section, as follows:

$$\overline{C}_{ij}^{I}(t) = \frac{Q_{IV} C_{ij-1}(l_{n-1}, t)}{Q_I}, \text{Section I, 1st column}$$

$$\overline{C}_{ij}^{III}(t) = \frac{Q_{II} C_{ij-1}(l_{n-1}, t) + Q_f C_{fi}}{Q_{III}}, \text{Section III, 1st column}$$

$$\overline{C}_{ij}^{\sec t}(t) = C_{ij-1}(l_{n-1}, t), \text{other} \tag{16}$$

By using Formulas (8)–(10), we can obtain the boundary numerate condition for the SMB system, involving the feed stream concentration ($C_f$) and the flow rate in each section ($Q$):

$$C_{ij}(n+1, k) + C_{ij}(n+1, k+1) = C_{ij}(n-1, k+1) + C_{ij}(n-1, k) \tag{17}$$

$$C_{ij}(2,k+1) + C_{ij}(2,k) - \frac{4hv_j}{D_i}(C_{ij}(1,k) - \overline{C}_{ij}^{\sec t}(k)) = C_{ij}(0,k+1) + C_{ij}(0,k) \qquad (18)$$

By setting $m = \frac{v^*s}{h}, n = \frac{Ds}{h^2}$, and from Formula (12), we can obtain Equation (19):

$$\begin{aligned}
&-(m+n)C_{ij}(l-1,k+1) + (1+FH_i+2n)C_{ij}(l,k+1) + (m-n)C_{ij}(l+1,k+1) \\
&= (m+n)C_{ij}(l-1,k) + (1+FH_i-2n)C_{ij}(l,k) - (m-n)C_{ij}(l+1,k) \qquad l \neq 1, n
\end{aligned} \qquad (19)$$

Substituting Equations (17) and (18) into Formula (12), we can obtain the next two boundary Equations (20) and (21):

$$\begin{aligned}
&(1+FH_i+2n)C_{ij}(1,k+1) - 2nC_{ij}(2,k+1) \\
&= (1+FH_i-2n-\tfrac{8m(m+n)}{n})C_{ij}(1,k) + 2nC_{ij}(2,k) + \tfrac{8m(m+n)}{n}\overline{C}_{ij}^{\sec t}(k)
\end{aligned} \qquad (20)$$

$$-2nC_{ij}(n-1,k+1) + (1+FH_i+2n)C_{ij}(n,k+1) = 2nC_{ij}(n-1,k) + (1+FH_i-2n)C_{ij}(n,k) \qquad (21)$$

Denote the matric

$$A = \begin{bmatrix}
1+FH_i+2n & -2n & 0 & \cdots & 0 \\
-(m+n) & 1+FH_i+2n & (m-n) & \cdots & 0 \\
\vdots & \vdots & \vdots & \vdots & \vdots \\
0 & \cdots & -(m+n) & 1+FH_i+2n & (m-n) \\
0 & \cdots & 0 & -2n & 1+FH_i+2n
\end{bmatrix}$$

$$B = \begin{bmatrix}
1+FH_i-2n-\frac{8m(m+n)}{n} & 2n & 0 & \cdots & 0 \\
(m+n) & 1+FH_i-2n & -(m-n) & \cdots & 0 \\
\vdots & \vdots & \vdots & \vdots & \vdots \\
0 & \cdots & (m+n) & 1+FH_i-2n & -(m-n) \\
0 & \cdots & 0 & 2n & 1+FH_i-2n
\end{bmatrix}$$

$$w(k) = \left( \tfrac{m^2}{m+n}\overline{C}_{ij}^{\sec t}(k) \quad 0 \quad \cdots \quad 0 \quad 0 \right)^T$$

The iterative equation can be obtained as:

$$AC_{ij}(k+1) = BC_{ij}(k) + w(k) \qquad (22)$$

*2.3. Stability Analysis*

In the discussion of the stability of the Crank–Nicolson method applied to the SMB system, it is important to consider the source of errors in the process of digitalizing partial differential equations (PDEs). Two types of errors commonly arise: truncation errors resulting from derivative approximation and discretization, and error amplification inherent in the numerical method itself. To estimate truncation errors, the Taylor error formula can be employed, which provides an approximation of the error introduced by the discretization and derivative approximations. To explore error amplification, it is necessary to closely observe the behavior of the finite difference method. Von Neumann stability analysis is a technique commonly used to measure error amplification in numerical methods. The stability of the method requires selecting an appropriate step size or time increment to ensure that the amplification factor, as measured using Von Neumann stability analysis, does not exceed 1 for the $A^{-1}B$. By carefully considering the truncation errors and conducting the Von Neumann stability analysis with an appropriate step size, we can assess the stability of the Crank–Nicolson method when applied to the specific equation of interest in the SMB system.

Let $C_{ij}^*(k)$ be the exact solution satisfying Formula (22) and let $y_{ij}(k)$ be the approximate solution obtained by calculation and satisfying $Ay_{ij}(k+1) = By_{ij}(k) + w(k)$; the difference between them is $e_{ij}(k) = y_{ij}(k) - C_{ij}^*(k)$, satisfying:

$$Ae_{ij}(k) = Ay_{ij}(k) - AC_{ij}^*(k) = By_{ij}(k-1) + w(k-1) - (BC_{ij}^*(k-1) + w(k-1)) \tag{23}$$
$$= Be_{ij}(k-1)$$

If $A$ is nonsingular, we can obtain the error iteration:

$$e_{ij}(k) = A^{-1}Be_{ij}(k-1) \tag{24}$$

In order to ensure that the error $e_{ij}(k)$ is not amplified, a spectral radius of $A^{-1}B$ must satisfy $\rho(A^{-1}B) < 1$.

**Theorem 1.** *If matrix A is strictly diagonally dominant and it satisfies* $|a_{ii}| > \sum\limits_{\substack{j=1 \\ j \neq i}}^{n} |a_{ij}|$, *(i = 1,*

*2, ..., n), then matrix A is nonsingular.*

**Proof.** If matrix $A$ is a singular matrix, then there is a non-zero vector $x$ that satisfies $Ax = 0$. Set $|x_1| = \max\{|x_1|, |x_2| \cdots |x_n|\}$ , so $|x_1| \neq 0$, we can obtain:

$$a_{11}x_1 + a_{12}x_2 + \cdots a_{1n}x_n = 0, a_{11} = -a_{12}\frac{x_2}{x_1} - \cdots a_{1n}\frac{x_n}{x_1}$$

$$|a_{11}| \leq |a_{12}||\frac{x_2}{x_1}| + \cdots |a_{1n}||\frac{x_n}{x_1}| \leq \sum_{j=2}^{n} a_{1j}$$

This contradicts the condition, so matrix A is nonsingular. $\square$

Now, let us review the matrix in the SMB iteration formula.
Because

$$A = \begin{bmatrix} 1 + FH_i + 2n & -2n & 0 & \cdots & 0 \\ -(m+n) & 1 + FH_i + 2n & (m-n) & \cdots & 0 \\ \vdots & \vdots & \vdots & \vdots & \vdots \\ 0 & \cdots & -(m+n) & 1 + FH_i + 2n & (m-n) \\ 0 & \cdots & 0 & -2n & 1 + FH_i + 2n \end{bmatrix}$$

$$B = \begin{bmatrix} 1 + FH_i - 2n - \frac{8m(m+n)}{n} & 2n & 0 & \cdots & 0 \\ (m+n) & 1 + FH_i - 2n & -(m-n) & \cdots & 0 \\ \vdots & \vdots & \vdots & \vdots & \vdots \\ 0 & \cdots & (m+n) & 1 + FH_i - 2n & -(m-n) \\ 0 & \cdots & 0 & 2n & 1 + FH_i - 2n \end{bmatrix}$$

Let $m = \frac{v^*s}{h} = O(s), n = \frac{Ds}{h^2} = O(s)$, represent the infinitesimals of the same order. Provided that the time step is sufficiently small, matrices $A$ and $B$ exhibit strict diagonal dominance, rendering them nonsingular. Therefore, we can establish the stability of the iterative process involved in calculating the SMB system using the Crank–Nicolson method. By ensuring that the time step is small enough, the strict diagonal dominance of matrices $A$ and $B$ guarantees their nonsingularity, reinforcing the stability of the iterative process employed in the computation of the SMB system using the Crank–Nicolson method.

**Theorem 2.** *If the space step (h) and time step (s) are both sufficiently small, $AC_{ij}(k+1) = BC_{ij}(k) + w(k)$ is stable.*

**Proof.** Hypothesis $\lambda$ is the eigenvalue of $A^{-1}B$ and $v$ is the corresponding eigenvector. Select $||v||_\infty = 1$, the dimension of Matrix A is $p \times p$. So, $|v_k| \leq 1, 1 \leq k \leq p$, whereby component index $v_l = 1$ is chosen. Indicators are divided into three situations:

(1)  When $2 \leq l \leq p - 1$. According to $A^{-1}Bv = \lambda v \Rightarrow Bv = \lambda Av$, so we can obtain the equation in component $l$:

$$(m+n)v_{l-1} + (1+FH-2n)v_l + (n-m)v_{l+1} = \lambda[-(m+n)v_{l-1} + (1+FH+2n)v_l + (m-n)v_{l+1}]$$

$$\text{If}|\lambda| = \frac{|(m+n)v_{l-1} + (1+FH-2n) + (n-m)v_{l+1}|}{|[-(m+n)v_{l-1} + (1+FH+2n) + (m-n)v_{l+1}]|} < 1$$

Because $m = \frac{v^*s}{h} = O(s), n = \frac{Ds}{h^2} = O(s)$, so

$$\forall \varepsilon > 0, \exists \delta, when\ s < \delta(h), can\ get |m| \leq \varepsilon, |n| \leq \varepsilon$$

For any $s$, select a space step $h$ that is small enough to have $n > m \Rightarrow h < \frac{D}{v^*}$. Hence, we obtain

$$(m+n)v_{l-1} + (n-m)v_{l+1} < 2n \tag{25}$$

For the split line over the fixed point (1, 1)

$$(m+n)v_{l-1} + (n-m)v_{l+1} = 2n$$

Rectangle $(v_{l-1}, v_{l+1}) \in [-1, 1] \times [-1, 1]$ (Figure 2) is under the split line, so it satisfies the inequality (25) except for $(1, 1)$, which is shown in Figure 2. Therefore, $|\lambda| \leq 1$ can be inferred.

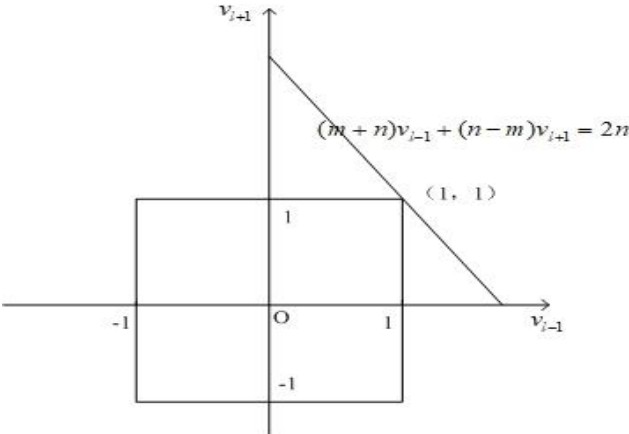

**Figure 2.** Space split line of $(v_{l-1}, v_{l+1})$.

Equality is established only if and only if $v_{l-1} = 1, v_{l+1} = 1$.

Now, it has been proven that if the components of the eigenvector meet this condition, contradictions arise for the previous component:

$$(m+n)v_{l-2} + (1+FH-2n)v_{l-1} + (n-m)v_l = [-(m+n)v_{l-2} + (1+FH-2n)v_{l-1} + (m-n)v_l]$$

where $v_{l-2} = 1$ can be obtained from it.

Any component can be obtained by the same principle, $v_k = 1, k = 1, \cdots p$.

But for the first component equation

$$(1 + FH - 2n - \frac{8m(m+n)}{n}) + 2n = (1 + FH + 2n) - 2n \Rightarrow \frac{8m(m+n)}{n} = 0$$

as $m = \frac{v^*s}{h} > 0, n = \frac{Ds}{h^2} > 0$, so $\frac{8m(m+n)}{n} > 0$, the statement contradicts the previous equation $\frac{8m(m+n)}{n} = 0$.

Thus, $|\lambda| < 1$.

(2)    When $l = 1$, it means that

$$(1 + FH - 2n - \frac{8m(m+n)}{n}) + 2nv_2 = \lambda[(1 + FH + 2n) - 2nv_2] \Rightarrow$$

$$\text{If } |\lambda| = \frac{(1 + FH - 2n - \frac{8m(m+n)}{n}) + 2nv_2}{(1 + FH + 2n) - 2nv_2} < 1$$

$$v_2 < 1 + \frac{2m(m+n)}{n^2}$$

Because $||v||_\infty = 1$, so $|\lambda| < 1$

(3)    When $l = p$, it means that

$$1 + FH - 2n + 2nv_{p-1} = \lambda[(1 + FH + 2n) - 2nv_{p-1}] \Rightarrow$$

$$\text{If } |\lambda| = \frac{1 + FH - 2n + 2nv_{p-1}}{[(1 + FH + 2n) - 2nv_{p-1}]} < 1$$

$$v_{p-1} < 1$$

When $v_{p-1} = 1 \Rightarrow \lambda = 1$.

Now, it has been proven that if the components of the eigenvector are in this situation, contradictions will occur.

$$(m+n)v_{l-2} + (1 + FH - 2n)v_{l-1} + (n-m)v_l = [-(m+n)v_{l-2} + (1 + FH - 2n)v_{l-1} + (m-n)v_l]$$

Like the first case, any component can be obtained by the same principle, $v_k = 1, k = 1, \cdots p$. Thus, $|\lambda| < 1$ and can be proven. $\square$

## 3. Simulation

### 3.1. Experimental Environment and Data

In the simulated digitization system, there are eight packed columns arranged in a 2-2-2-2 model. The initial parameters of the system are provided in Table 2. The time step is set to 0.1 s, and there are 50 spatial points for each column.

The research experiments were conducted using the MATLAB R2016a software on a PC equipped with an Intel Core i7-3770K 3.53 GHz processor and 16 GB RAM. The experimental data were generated based on simulated experiments, resulting in a data volume of 70 MB. The separation process of SMB without any control conditions is shown in Figure 3.

**Table 2.** The standard parameters for the separation.

| Parameter | Value | Parameter | Value |
|---|---|---|---|
| $L(\text{cm})$ | 25 | $C_{f,i}(\text{gL}^{-1})$ | 5 |
| $d(\text{cm})$ | 0.46 | $\theta(\text{min})$ | 3 |
| $H_A$ | 0.001 | $Q_I(\text{cm}^3\text{min}^{-1})$ | 6.75 |
| $H_B$ | 0.45 | $Q_{II}(\text{cm}^3\text{min}^{-1})$ | 6.6 |
| $D_A(\text{cm}^2\text{min}^{-1})$ | 0.2 | $Q_{III}(\text{cm}^3\text{min}^{-1})$ | 7 |
| $D_B(\text{cm}^2\text{min}^{-1})$ | 1.265 | $Q_{IV}(\text{cm}^3\text{min}^{-1})$ | 2 |
| $\varepsilon$ | 0.8 | spatial number | 50 |

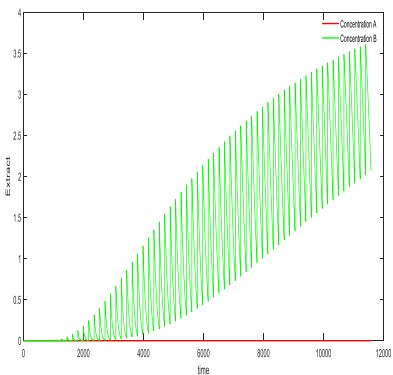 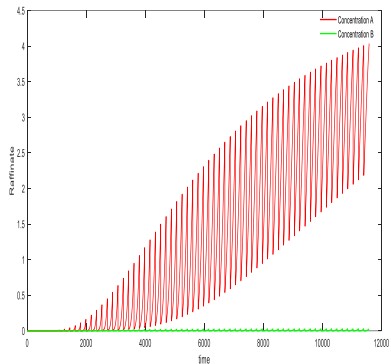

**Figure 3.** Concentration of the SMB process on extract and raffinate.

### 3.2. Sensitivity Analysis of Purity Flow Rates to Find Local Monotonic Intervals

In Figure 4, $Q_I$ changes from 7.32 to 7.5, and when $Q_{II} = 6.96$, $Q_{III} = 8.4$ and $Q_{IV} = 2$ remain unchanged and the switch time is 180 s. The purity of extract solution for material B decreases with the increase in the flow rate in Zone I. However, the concentration of raffinate for material A remains almost unchanged.

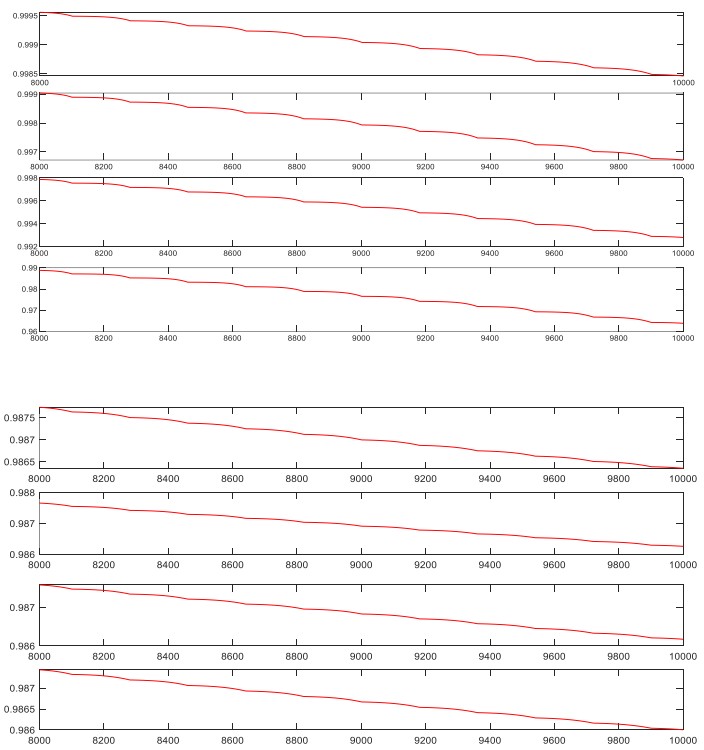

**Figure 4.** The influence of Zone I's flow rate on purity.

In Figure 5, $Q_{II}$ changes from 6.92 to 7.1, and when $Q_I$ = 7.14, $Q_{III}$ = 8.4 and $Q_{IV}$ = 2, and the switch time is 180 s. The purity of the extract solution of material B remains almost unchanged as the flow rate of Zone II increases, but the purity of raffinate for material A decreases with an increase in the flow rate of $Q_{II}$.

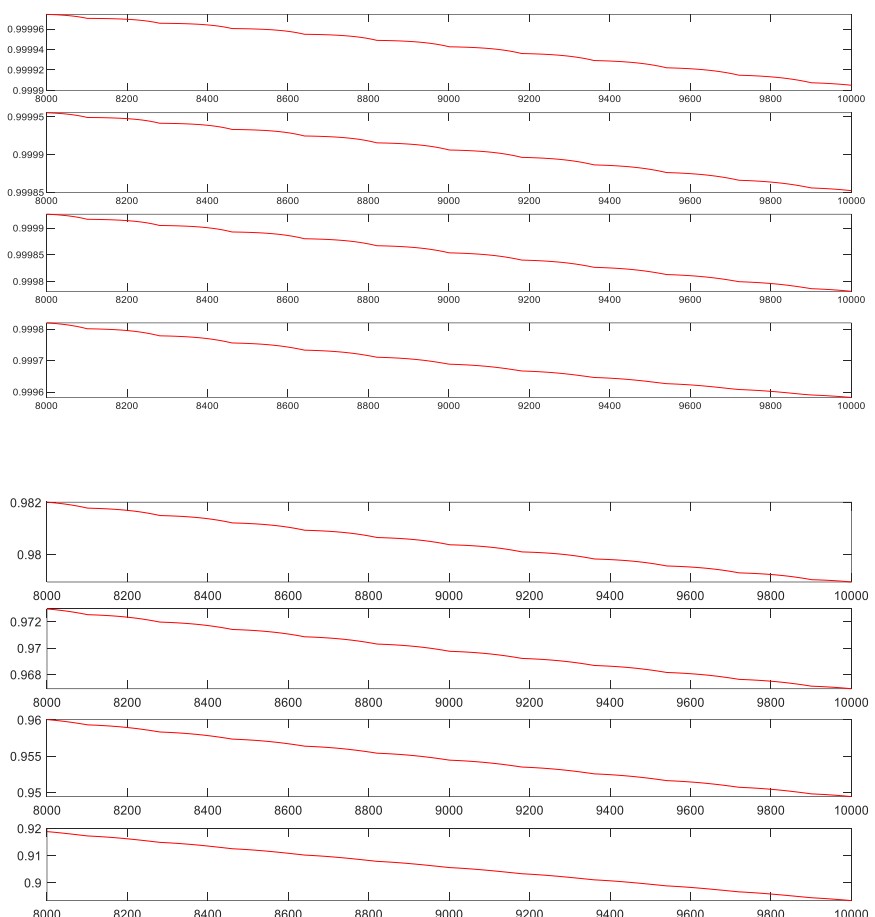

**Figure 5.** The influence of Zone II's flow rate on purity.

Similarly, upon conducting a sensitivity analysis on flow velocity in zones III and IV, $Q_{III}$ changes from 8.4 to 12, and when $Q_I$ = 6.96, $Q_{II}$ = 7.2 and $Q_{IV}$ = 2, and the switch time is 180 s. The purity of extract solution of material B decreases with an increase in the flow of Zone III, and the purity of raffinate of material A decreases with the increase in the flow of Zone III. But the decreasing tendency is very small, so it is better for fine tuning. However, whether it is material A or material B, the change amount is relatively small, so the flow rate in this area is suitable for fine-tuning control. The sensitivity analysis of flow velocity in Zone IV is similar to that in Zone III, so it is also suitable for fine tuning the control effect.

## 4. Smart Controller Design

As mentioned before, SMB is a very complex nonlinear system. The control of a multi-column SMB system cannot be accomplished independently by one controller. Each feed hole must be equipped with an independent controller for control. The purpose of chromatographic separation can only be achieved through a precise and appropriate feed speed control of the flow rate of the material to be separated. Therefore, how to design a smart controller with learning and self-adjustment abilities is another aim of this study in SMB control applications. In view of this, the smart controller designed in this research combines the characteristics of fuzzy control and neural network (NN) control. It is an NN-like fuzzy controller. It adopts the concept of "ANFIS: adaptive-network-based fuzzy inference system" proposed by J-S.R. Jang as the controller architecture [27,28].

### 4.1. Fuzzy Rule Control and Hierarchical Design

First, several symbolic meanings in the control are defined as follows:

$$\Delta e(k) = e(k) - e(k-1) \tag{26}$$

$$\Delta e(k-1) = e(k-1) - e(k-2) \tag{27}$$

$$e(k) = desired\ output - y(k) \tag{28}$$

In the SMB control system, the fuzzy input variables are selected as error ($e(k)$) and error change ($\Delta e(k)$); the formula is as follows:

$$e_1(k) = B \text{ material desired output} - C_{E,B} \tag{29}$$

$$e_2(k) = A \text{ material desired output} - C_{R,A} \tag{30}$$

$$\Delta e_1(k) = e_1(k) - e_1(k-1) \tag{31}$$

$$\Delta e_2(k) = e_2(k) - e_2(k-1) \tag{32}$$

In the smart controller mechanism, the control output is the liquid flow rate increment. The selected input fuzzy variables are error and error change. The fuzzy system defines five linguistic variable values for error and error change, respectively: NB, NS, ZE, PS, and PB. The graph of the membership function is shown in Figure 6.

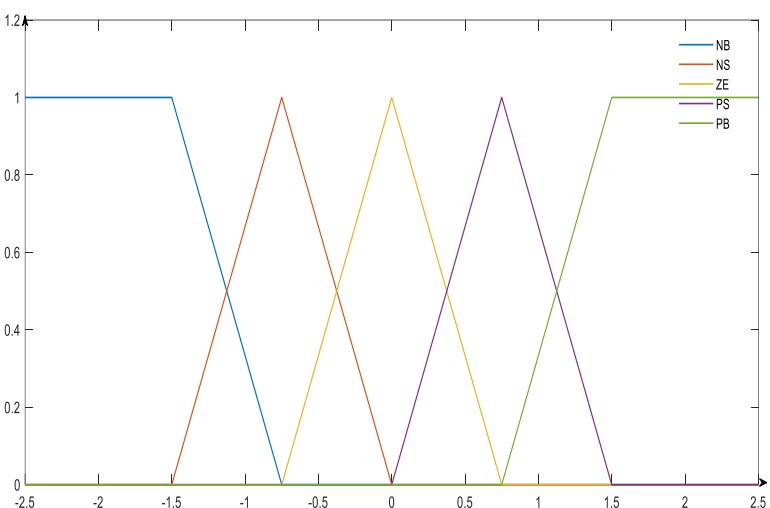

**Figure 6.** Membership function of e(k) and $\Delta$e(k).

In the subsequent SMB control process, the increments of the flow rate $Q_I$ in region I, $Q_{II}$ in region II, and $Q_{III}$ in region III are chosen as three independent defuzzification output variables denoted as $\Delta Q_i$ ($i = 1, 2, 3$), respectively. The membership function of $\Delta Q_i$ is shown in Figure 7.

Since purity decreases with an increase in flow rate, the values of ($C_1, C_2, C_3, C_4, C_5$) are set as {$-0.15, -0.1, 0, 0.1, 0.15$} for $\Delta Q_1$, {$-0.006, -0.004, 0, 0.0.004, 0.006$} for $\Delta Q_2$, and {$-0.08, -0.05, 0, 0.05, 0.08$} for $\Delta Q_3$. When conducting sensitivity analysis near the flow rate region, it is necessary to adjust the rule table to reflect the decrease in purity with the increasing flow rate. Thus, the rule table was developed as listed in Table 3.

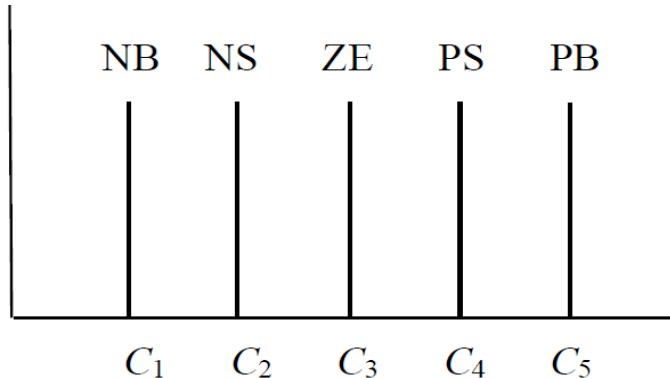

**Figure 7.** Membership function of $\Delta Q_i$ ($i$ = 1, 2, 3).

**Table 3.** Rule table of $\Delta Q_i$ ($i$ = 1, 2, 3).

| $\Delta e$ \ e | NB | NS | ZE | PS | PB |
|---|---|---|---|---|---|
| NB | PB | PB | PB | PS | NB |
| NS | PB | PS | PS | ZE | NB |
| ZE | PB | PS | ZE | NS | NB |
| PS | PB | ZE | NS | NS | NB |
| PB | PB | NS | NB | NB | NB |

Although the system is a multiple-input and multiple-output (MIMO) system, there is no coupling between the velocity in the first region and the velocity in the second region. Additionally, the flow velocity in the third region can be fine-tuned independently. This allows for the execution of the control process sequentially in the design control process.

In the preceding part of the reasoning, the Mamdani operator, basic complementary operation, and Mamdani fuzzy reasoning rule are utilized. The velocity in Zone I primarily affects the purity of the extract material B, while the velocity in Zone II mainly influences the purity of the raffinate material A. The effect of Zone IV's velocity is similar to that of Zone III; hence, Zone III's velocity is selected as the control variable. The action diagrams of the three independent controllers are shown in Figure 8.

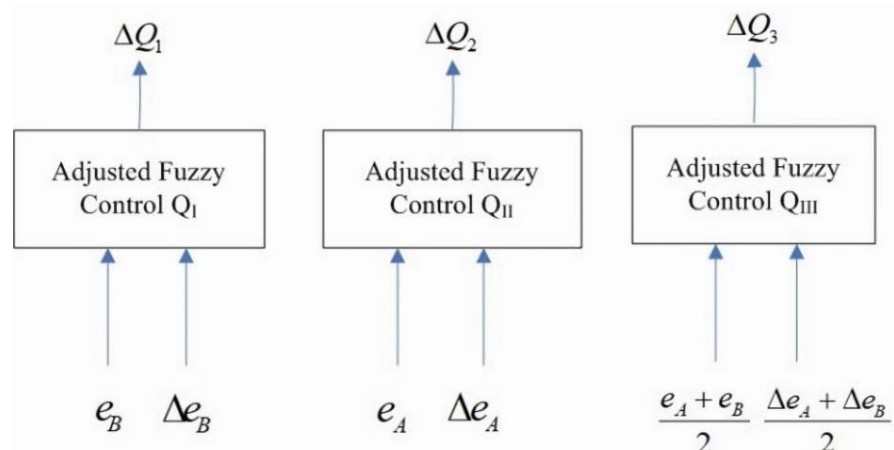

**Figure 8.** Three independent controllers.

*4.2. NN-like Fuzzy Controller Framework*

The NN-like fuzzy controller is divided into five layers, as shown in Figure 9. The functions of each layer are described below.

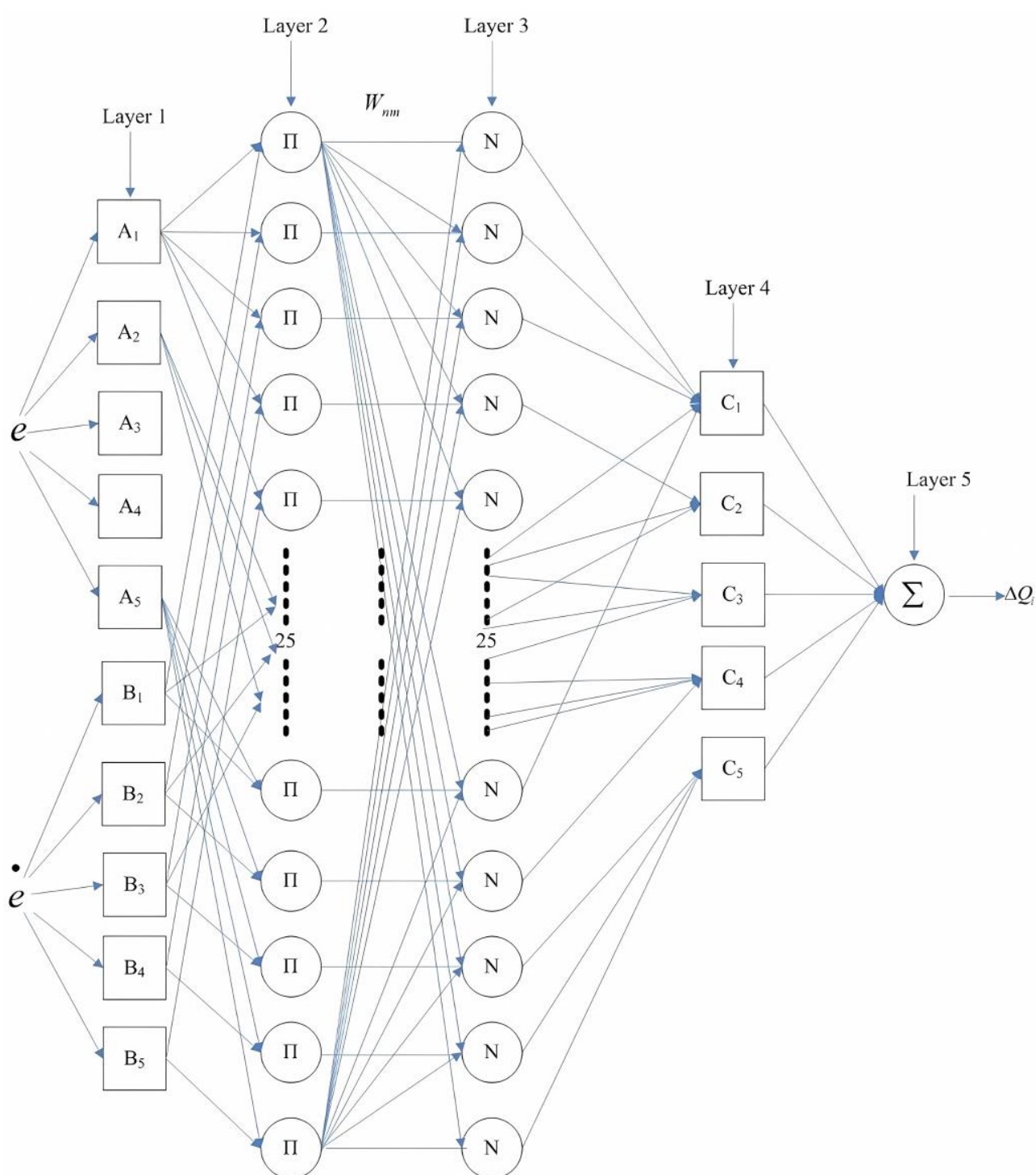

**Figure 9.** NN-like fuzzy controller framework.

　　　Layer 1: This layer calculates the values of the fuzzy membership function corresponding to the input variables, including the error and error change. $u_{A_i}$ represents the error membership function values. $u_{B_i}$ represents the error change membership function value.

Layer 2: In this layer, the starting strength of rule base is calculated according to the fuzzy rule base. $Z_i$ represents the starting strength:

$$\begin{aligned}
Z_1 &= u_{A_1} \times u_{B_1} \\
Z_2 &= u_{A_1} \times u_{B_2} \\
&\vdots \\
Z_{24} &= u_{A_5} \times u_{B_4} \\
Z_{25} &= u_{A_5} \times u_{B_{25}}
\end{aligned} \tag{33}$$

Layer 3: The third layer performs the regularization operation of fuzzy start-up intensity. $W_{nm}$ stands for connection weight, and $net_n$ represents regularization output fuzzy start-up intensity:

$$net_n = \frac{Z_n \times W_{nn}}{\sum\limits_{m=1}^{25} Z_m \times W_{nm}}, n = 1, 2, 3 \cdots 25 \tag{34}$$

Layer 4: This layer calculates the startup of the IF part after the regularization operation of the fuzzy rules. $net_n$ is the input of this layer, $O_{4,n}$ is the output of this layer, and $C_{n_k}$ indicates that the layer is a shared weight:

$$O_{4,n} = net_n \times C_{n_k} \tag{35}$$

$$C_1 = C_{NB}, C_2 = C_{NS}, C_3 = C_{ZE}, C_4 = C_{PS}, C_5 = C_{PB} \tag{36}$$

Layer 5: This layer calculates the final control output of the neuron:

$$\Delta u = \sum_{n=1}^{25} net_n \times C_{n_k} \tag{37}$$

By updating weights using back-propagation algorithms similar to neural networks, it can obtain the updating formulas, as follows:

$$\text{Set loss function}: \ E = \frac{1}{2}(desired - y)^2 \tag{38}$$

$$\delta_o = -\frac{\partial E}{\partial y}\frac{\partial y}{\partial u}\frac{\partial u}{\partial \Delta u} = (desire - y) \times \text{sgn}(\frac{y(t) - y(t-1)}{u(t) - u(t-1)}) \times 1 \tag{39}$$

$$\frac{\partial E}{\partial C_1} = \frac{\partial E}{\partial y}\frac{\partial y}{\partial u}\frac{\partial u}{\partial \Delta u}\frac{\partial \Delta u}{\partial C_1} = -\delta_o \times (net_1 + net_2 + net_3 + net_6 + net_{11} + net_{16} + net_{21}) \tag{40}$$

$$\frac{\partial E}{\partial C_2} = \frac{\partial E}{\partial y}\frac{\partial y}{\partial u}\frac{\partial u}{\partial \Delta u}\frac{\partial \Delta u}{\partial C_2} = -\delta_o \times (net_4 + net_7 + net_8 + net_{12}) \tag{41}$$

$$\frac{\partial E}{\partial C_3} = \frac{\partial E}{\partial y}\frac{\partial y}{\partial u}\frac{\partial u}{\partial \Delta u}\frac{\partial \Delta u}{\partial C_3} = -\delta_o \times (net_9 + net_{13} + net_{17}) \tag{42}$$

$$\frac{\partial E}{\partial C_4} = \frac{\partial E}{\partial y}\frac{\partial y}{\partial u}\frac{\partial u}{\partial \Delta u}\frac{\partial \Delta u}{\partial C_4} = -\delta_o \times (net_{14} + net_{18} + net_{19} + net_{22}) \tag{43}$$

$$\frac{\partial E}{\partial C_5} = \frac{\partial E}{\partial y}\frac{\partial y}{\partial u}\frac{\partial u}{\partial \Delta u}\frac{\partial \Delta u}{\partial C_5} = -\delta_o \times (net_5 + net_{10} + net_{15} + net_{20} + net_{23} + net_{24} + net_{25}) \tag{44}$$

$$\Delta C_n = \eta \times \frac{\partial E}{\partial C_n} \tag{45}$$

where $\eta$ is learning rate:

$$\frac{\partial E}{\partial W_{nm}} = \frac{\partial E}{\partial y} \frac{\partial y}{\partial u} \frac{\partial u}{\partial \Delta u} \frac{\partial \Delta u}{\partial net_n} \frac{\partial net_n}{\partial W_{nm}} \tag{46}$$

$$\delta_n = \delta_o \times C_{n_k} \tag{47}$$

$$\frac{\partial E}{\partial W_{nm}} = -\delta_n \times \frac{Z_n \times \sum\limits_{m=1}^{25} Z_m \times W_{nm} - Z_m \times Z_n \times W_{nn}}{(\sum\limits_{m=1}^{25} Z_m \times W_{nm})^2}, m = n \tag{48}$$

$$\frac{\partial E}{\partial W_{nm}} = -\delta_n \times \frac{-Z_m \times Z_n \times W_{nn}}{(\sum\limits_{m=1}^{25} Z_m \times W_{nm})^2}, m \neq n \tag{49}$$

$$\Delta W_{nm} = \eta \times \frac{\partial E}{\partial W_{nm}} \tag{50}$$

## 5. SMB Control Experiments

### 5.1. Purity Control Result

To verify the applicability of the NN-like fuzzy controllers in SMB control, we conducted several experiments on the purity control of the SMB system. Figures 10–12 show the effects of material separation results for materials *A* and *B* located at the outlets of the extraction (material *B*) and raffinate (material *A*).

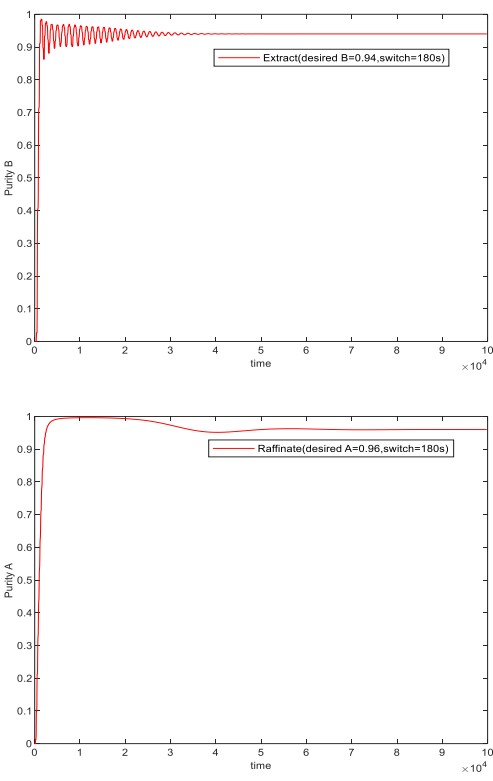

**Figure 10.** Purity control of the first experiment (switch time = 180 s) (actual *B* = 94%, desired *B* = 94%, actual *A* = 96%, desired *A* = 96%).

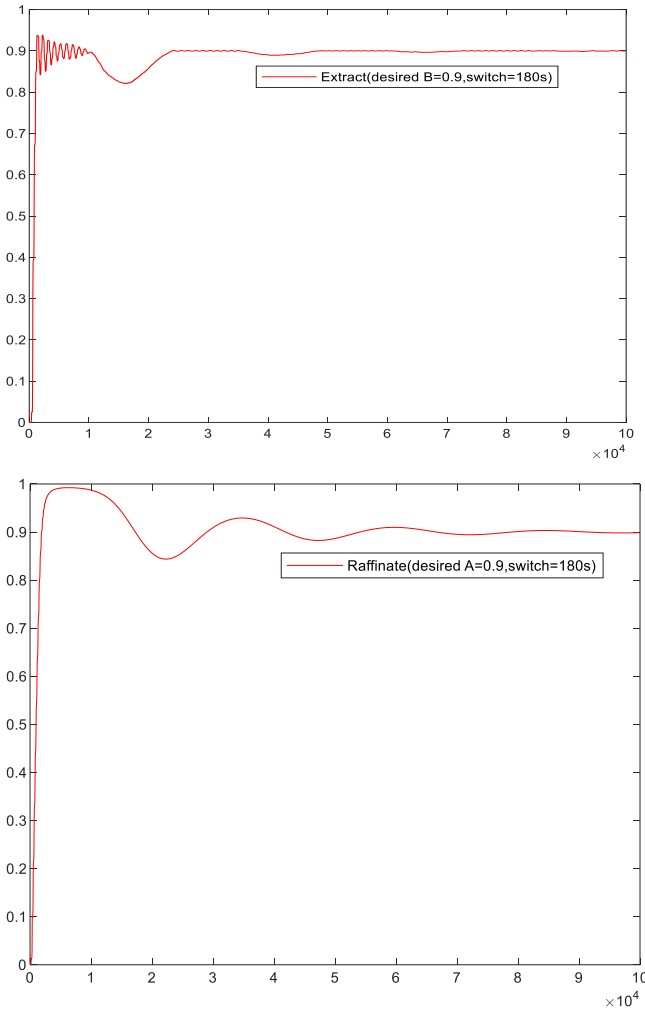

**Figure 11.** Purity control of the second experiment (switch time = 180 s) (actual *B* = 89.98%, desired *B* = 90%, actual *A* = 89.86%, desired *A* = 90%).

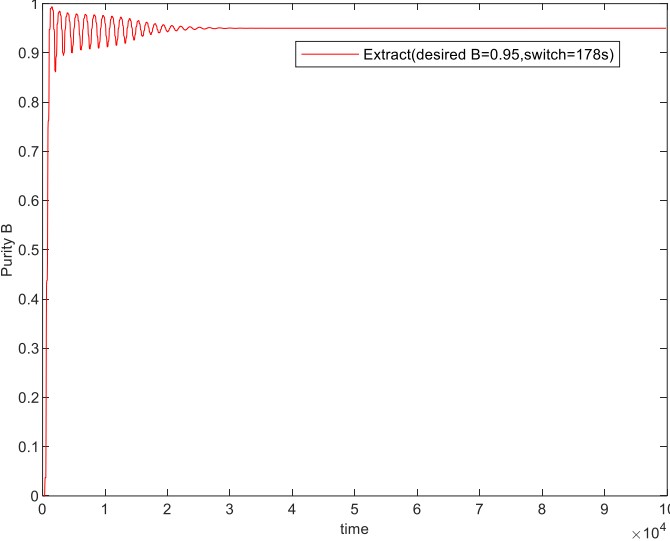

**Figure 12.** *Cont*.

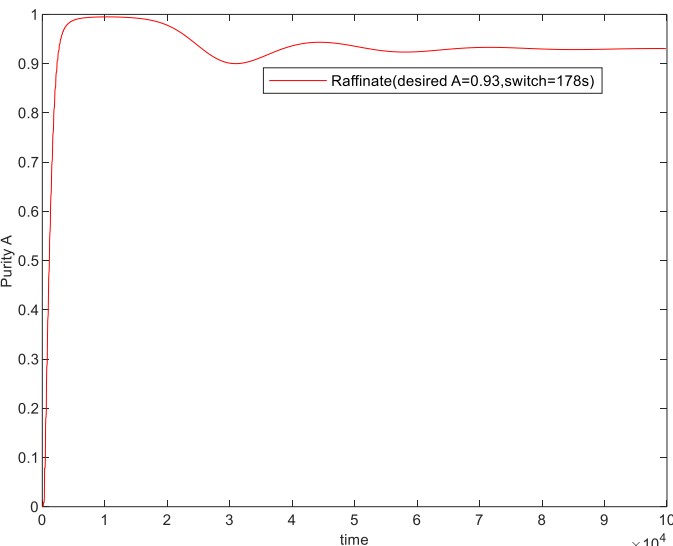

**Figure 12.** Purity control of the fourth experiment (switch time = 178 s) (actual *B* = 95%, desired *B* = 95%, actual *A* = 93.07%, desired *A* = 93%).

From Figures 10–12, it is evident that the NN-like fuzzy controller performs with high accuracy in controlling the purity of the extraction and raffinate outlet materials. The steady-state error in controlling the purity of the extraction solution's outlet material and the raffinate solution's outlet material is nearly negligible. The entire control process is quite stable, especially for the purity of raffinate outlet material A with higher sensitivity.

*5.2. Controller Comparison*

In order to demonstrate that the NN-like fuzzy controller has excellent robustness and adaptability in SMB control compared with the traditional fuzzy controller, we also conducted an experimental comparison of the two controllers.

From Figures 13 and 14, it is evident that the NN-like fuzzy controller outperforms the traditional fuzzy controller. In comparison to the traditional fuzzy controller, the NN-like fuzzy controller exhibits superior performance. Specifically, when considering the purity of substance *B* at the outlet of extraction, the traditional fuzzy controller shows steady-state errors and fluctuations, whereas the NN-like fuzzy controller maintains stability without any steady-state errors throughout. There is no significant difference in the control effect for the concentration of substance at the raffinate outlet. However, when the switching time is changed to 178 s, the traditional fuzzy controller displays ill-conditioned characteristics. This is because it does not adjust the membership function of the force intensity of the control force or the center value of the force output. On the other hand, the NN-like fuzzy controller remains robust and unaffected by the switching time disturbance.

In Figures 15–17, both controllers demonstrate the ability to maintain stability under the disturbance of adsorbent parameters and inlet concentration variations. However, the NN-like fuzzy controller exhibits higher stability compared to the traditional fuzzy controller. Furthermore, when faced with the disturbance of switching time with high sensitivity, the traditional fuzzy controller once again exhibits ill-conditioned features, while the NN-like fuzzy controller remains robust. Therefore, the NN-like fuzzy controller proves to be a robust and adaptive solution for highly complex and sensitive SMB systems.

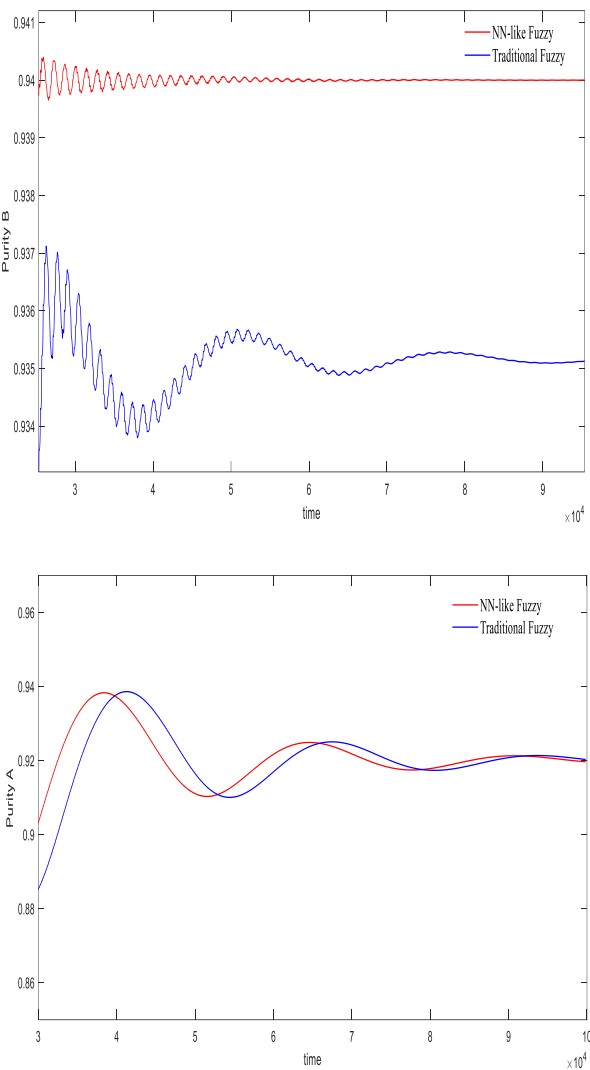

**Figure 13.** Comparison of two controllers (desired *B* = 94%, desired *A* = 92%, switch time = 180 s).

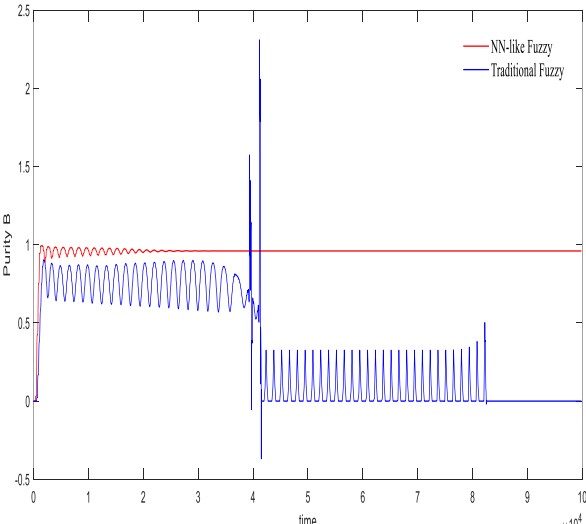

**Figure 14.** *Cont*.

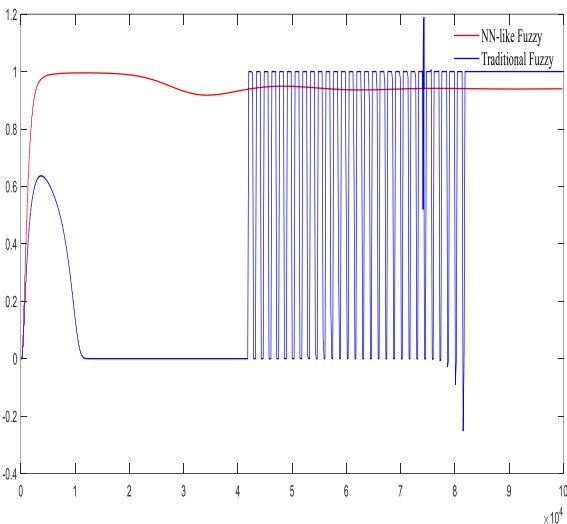

**Figure 14.** Comparison of two controllers (desired *B* = 96%, desired *A* = 94%, switch time = 178 s).

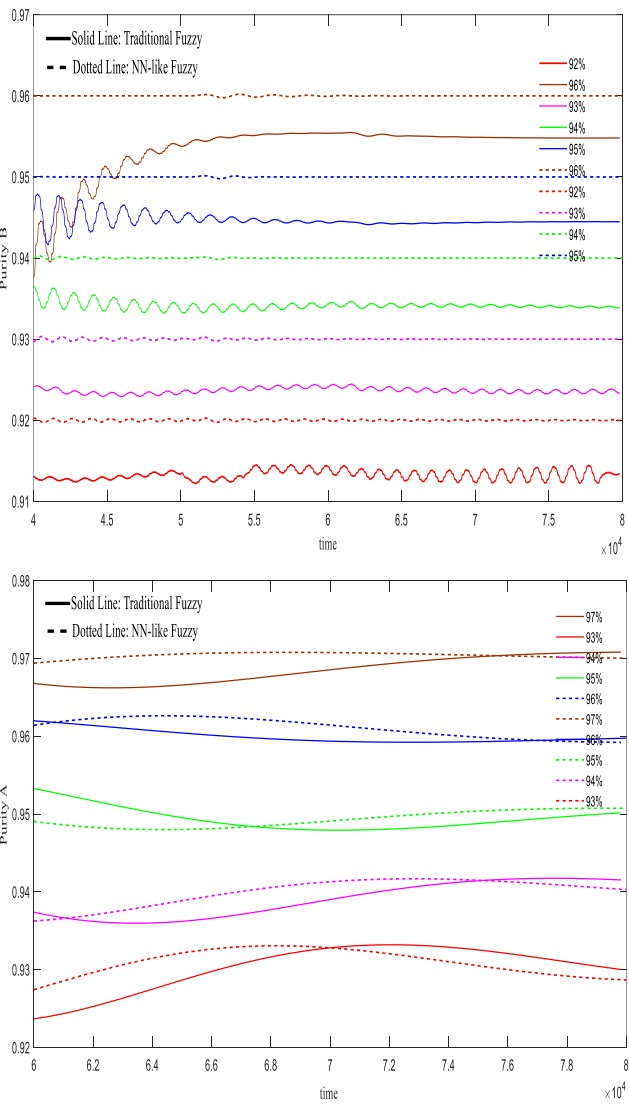

**Figure 15.** Comparison of two controllers under the disturbance of adsorbent parameters $H_A = 0.01 \rightarrow 0.03$.

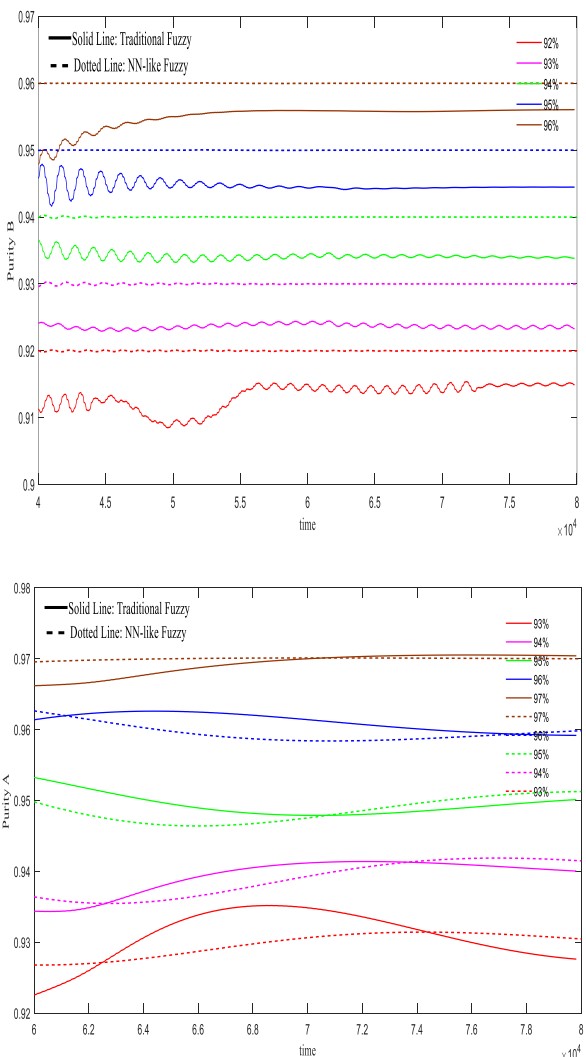

**Figure 16.** Comparison of two controllers under the disturbance of feed port concentration $C_f = 4.5 \rightarrow 5.2$.

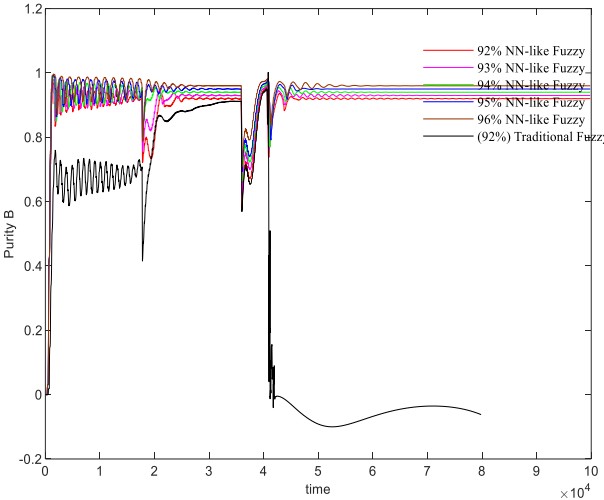

**Figure 17.** *Cont.*

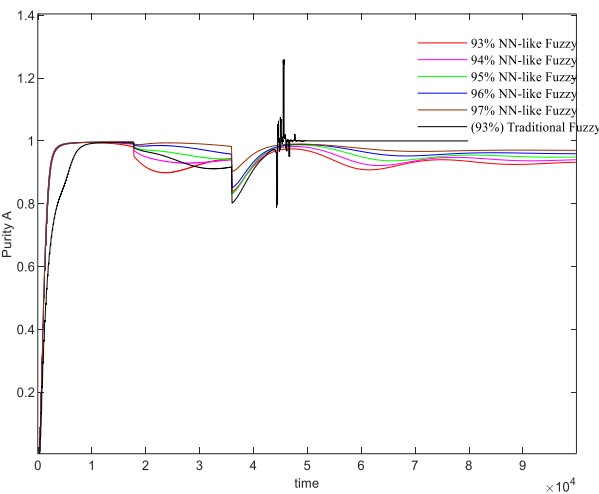

**Figure 17.** Comparison of two controllers under the disturbance of switch time $\theta = 178s \rightarrow 182s$.

## 6. Discussion

From the experimental results, it is evident that the NN-like fuzzy controller outperforms the traditional fuzzy controller in terms of steady-state error, control stability, and response to parameter perturbations, particularly for the residual liquid outlet. However, the downside of the NN-like fuzzy controller is the significantly higher computational burden compared to the traditional fuzzy controller. Given the current hardware structure and the neural network's layer and architecture, it can adequately support the computational load. Therefore, the NN-like fuzzy controller holds immense practical value and is expected to perform exceptionally well in practical applications.

## 7. Conclusions

This paper delves into the digitalization of the SMB process by employing the Crank–Nicolson method to construct a dynamic model. The stability and convergence of this method for solving partial differential equations (PDEs) within the SMB process have been empirically validated. An integral component of our investigation involves a comprehensive sensitivity analysis of regional flow velocity in the SMB system, conducted through computer simulations. By scrutinizing the impact of varied regional velocities on the purity of both extract and raffinate outlets, our research identifies a monotone region of purity. Leveraging insights from this sensitivity analysis and guided by dynamic principles, we have formulated fuzzy control rules. These rules are further enhanced by the integration of a dynamic learning adjustment function, known as the NN-like fuzzy controller.

Our results indicate that the neural NN-like fuzzy controller demonstrates exceptional precision in regulating the purity of outlet streams with minimal steady-state errors. The controller also demonstrates remarkable robustness and adaptability in coping with disturbances resulting from fluctuations in system parameters, such as adsorbent properties, feed port concentration, and switching time.

In comparison to the conventional fuzzy controller, the NN-like fuzzy controller not only eradicates steady-state errors, but also showcases superior robustness and adaptability in the presence of variations and disturbances in diverse parameters. Consequently, the NN-like fuzzy controller emerges as an optimal choice for controlling highly complex SMB systems that are sensitive to dynamic parameter changes. This study contributes valuable insights to the field, offering superior alternative solutions with widespread practical applications in the industry.

**Author Contributions:** Conceptualization, R.-C.H. and C.-F.X.; methodology, C.-F.X.; software, C.-F.X.; validation, R.-C.H. and C.-F.X.; formal analysis, R.-C.H.; resources, R.-C.H.; data curation, C.-F.X., H.Z.; writing—original draft preparation, C.-F.X.; writing—review and editing, R.-C.H.; visualization,

H.Z., C.-F.X.; supervision, R.-C.H.; project administration, R.-C.H. All authors have read and agreed to the published version of the manuscript.

**Funding:** This research received no external funding.

**Data Availability Statement:** To request the corresponding research paper data for experimental simulation, please submit your application via the following email address: liurangzhu@163.com.

**Acknowledgments:** This research was funded by the National Natural Science Foundation of China (under Grant 62071123, 61601125), by Natural Science Foundation of Fujian Province of China (number: 2023J011117), by the Fujian Province Education Hall Youth Project (number: JAT220258), by the Fujian Natural Science Foundation Project (number: 2019J01887).

**Conflicts of Interest:** The authors declare no conflict of interest.

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
