# Peer review of "Application of Intelligent Control in Chromatography Separation Process"

_processes, doi:10.3390/pr11123443_

Round 1

Reviewer 1 Report

Comments and Suggestions for Authors

The manuscript deals with an improvement of the existing approaches to theoretical modelling chromatographic separation processes. Undoubtedly, it has a certain practical merit for various industrial applications and can be of interest for the Processes audience.  However, in my opinion, the mathematical model must be proved by the corresponding experimental data. I could not find any tests demonstrating an adequacy of the model on real chromatographic data and comparison with other existing approaches to the simulation of chromatographic separations. In this situation, I'm not convinced that the presented study is valuable and not speculative. The authors must clearly show the advantages of their model and its applicability to the practice. 

Minor comment. The quality of figures is too low and does not allow their perception. 

Comments on the Quality of English Language

Minor editing of English language required

Author Response

Thank the reviewer for the precious suggestions, according to your requirements, introduce some changes as follow:

1.Reviewer Question 1:In my opinion, the mathematical model must be proved by the corresponding experimental data. I could not find any tests  demonstrating an adequacy of the model on real chromatographic data and comparison with other existing approaches to the simulation of chromatographic separations.

Answer:(1)I have carefully reviewed the relevant literature, and the vast majority of academic papers start with computer simulation. Most of the experiments in these papers involve inputting relevant parameters into a computer and then conducting simulations. Comparing the obtained results with actual data does not directly apply the controller to the system. (2)If actual experiments are to be conducted, the cost of different materials will vary, and most of them will greatly increase the experimental cost. More importantly, the cost of implementing the hardware circuit of the controller is very large, and it also involves different pipe column structures and other issues. (3)This paper mainly simulates the process of moving beds from the perspective of the academic community, and designs a neural fuzzy controller that can adaptively adjust. The entire simulation results are based on the separation simulation, and the concentration changes in the process are consistent with reality. That is to say, the mathematical model is close to reality. Based on this, the effect of the newly designed controller is observed, and its superiority is discovered, providing corresponding reference for the industry, If the industry realizes its circuit structure and applies it to practical systems, I think this is also the value of this paper.

2.Reviewer Question 2:The quality of figures is too low and does not allow their perception.

Answer: Thank you again for your suggestions to me. I have reprocessed almost all the images in the paper, enhancing their pixels to make them look clearer. And some images that are not important to the entire research framework were removed, making the structure of the paper more refreshing.

3. All modified parts in the article are marked in red.

Thank you very much again for your valuable feedback. Your feedback is very useful for my academic research.

Reviewer 2 Report

Comments and Suggestions for Authors

In accordance with the principles of publishing research articles common to all fields of science, the manuscript should contain the following sections: introduction (relevance, novelty, previous work of other authors in this field), results, discussion (comparison of results with previous data and data of other authors), conclusions (briefly and succinctly).

The content of the submitted manuscript is interesting, but the form, in my opinion, needs correction. I believe it will not be difficult for the authors to change their manuscript in accordance with the common scientific style of the article.

Author Response

Thank the reviewer for the precious suggestions, according to your requirements, introduce some changes as follow:

1.Reviewer Question 1: In accordance with the principles of publishing research articles common to all fields of science, the manuscript should contain the following sections: introduction (relevance, novelty, previous work of other authors in this field), results, discussion (comparison of results with previous data and data of other authors), conclusions (briefly and succinctly).

Answer:  We carefully reviewed the manuscript, which includes an introduction, literature review, mathematical model, simulation, controller design, experimental results and comparisons, as well as conclusions. For the discussion section, we mainly focus on explaining the experimental results and comparisons, as well as the conclusion section.

2.Reviewer Question 2: The content of the submitted manuscript is interesting, but the form, in my opinion, needs correction.

Answer:First of all, thank you very much for your recognition of our research. For the manuscript revisions, we have rewritten the abstract, introduction, and conclusion sections, removing some parts that have little impact on the overall framework, making it look fresh and concise. For the graphic part, we have almost completely reprocessed it to make its pixels higher and clearer.

  1. All modified parts in the article are marked in red.

Thank you very much again for your valuable feedback. Your feedback is very useful for my academic research.

Reviewer 3 Report

Comments and Suggestions for Authors

The manuscript "Application of Intelligent Control in Chromatography Separation Process" has an important and actual subject of the research field!

The title is promising but the manuscript must be upgraded.

The Introduction and Literature sections can be can be presented together.

Some valuable references were lost!

The mathematical presentation is very extensive (77 equations) and should be reconsidered at least in part. For example, the passing of a model in the additional material and in the article should be treated as a black box with input and output parameters!

Many of the figures are difficult to read or have very poor graphic quality (example figs. 1,3-8, 10-18).

The conclusion must redone.

The abstract must be more concise.

Author Response

Thank the reviewer for the precious suggestions, according to your requirements, introduce some changes as follow:

1.Reviewer Question 1:The abstract must be more concise.

Answer: We have rewritten the abstract to make it more concise. The abstract has been rewritten as follows: 

Chromatographic separation plays a pivotal role in the manufacturing of chemical products and biopharmaceuticals. This technique exploits differences in distribution between stationary and mobile phases to separate mixtures, impacting final product quality. Simulated Moving Bed (SMB) technology, recognized for continuous feed, enhances efficiency, increasing production capacity while reducing solvent and water consumption. Despite its complexity in controlling variables like flow rates and valve switching times, traditional control theories fall short. This study introduces an intelligent fuzzy controller resembling an approximate neural network (NN) for SMB control. Simulation results demonstrate the controller's effectiveness in achieving de-sirable outcomes for the SMB system.

2.Reviewer Question 2: The conclusion must redone.

Answer: We have rewritten the conclusion as follows:

This paper delves into the digitalization of the SMB process by employing the Crank-Nicolson method to construct a dynamic model. The stability and convergence of this method for solving partial differential equations (PDEs) within the SMB process have been empirically validated. An integral component of our investigation involves a comprehensive sensitivity analysis of regional flow velocity in the SMB system, conducted through computer simulations. By scrutinizing the impact of varied regional velocities on the purity of both extract and raffinate outlets, our research identifies a monotone region of purity. Leveraging insights from this sensitivity analysis and guided by dynamic principles, we have formulated fuzzy control rules. These rules are further enhanced by the integration of a dynamic learning adjustment function, known as the NN-like fuzzy controller.

Our findings demonstrate that the NN-like fuzzy controller exhibits remarkable accuracy in regulating the purity of outlet streams, minimizing steady-state errors. Notably, it displays robustness and adaptability when faced with disturbances stemming from fluctuations in system parameters, including adsorbent properties, feed port concentration, and switching time.

In comparison to the conventional fuzzy controller, the NN-like fuzzy controller not only eradicates steady-state errors but also showcases superior robustness and adaptability in the presence of variations and disturbances in diverse parameters. Consequently, the NN-like fuzzy controller emerges as an optimal choice for controlling highly complex SMB systems that are sensitive to dynamic parameter changes. This study contributes valuable insights to the field, offering a practical and effective solution for enhancing the control of SMB processes.

3.Reviewer Question 3:Many of the figures are difficult to read or have very poor graphic quality (example figs. 1,3-8, 10-18).

Answer: Thank you again for your suggestions to me. I have reprocessed almost all the images in the paper, enhancing their pixels to make them look clearer. And some images that are not important to the entire research framework were removed, making the structure of the paper more refreshing.

4.Reviewer Question 4: The mathematical presentation is very extensive (77 equations) and should be reconsidered at least in part.

Answer: According to the requirements, we have reorganized the equations and removed some equations that have little impact on the overall architecture. The main body retains the discretization process of the simulated moving bed model and the learning process of the adaptive fuzzy neural network.

5.Reviewer Question 5: Some valuable references were lost.

Answer: For references, we adhere to the principle of seeking truth from facts and have indeed searched for a large number of literature. However, we have found that there are not many literature related to our research. If you have any valuable suggestions for references, we are very willing to read them to enhance our academic foundation.

6. All modified parts in the article are marked in red.

Thank you very much again for your valuable feedback. Your feedback is very useful for my academic research.

Round 2

Reviewer 1 Report

Comments and Suggestions for Authors

The authors have explained their position regarding practical applications of the developed approach and made some important amendments in the manuscript. Although I do not agree that theoretical calculations do not need practical testing, I admit that the manuscript has a certain value and recommend it for publication in its current form.

Comments on the Quality of English Language

Only minor corrections are required.

Author Response

Thank you again  for the precious suggestions, according to your requirements, introduce some changes as follow:

1.Reviewer Question 1: Only minor corrections are required

Answer:  I have done some polishing on the conclusion section and highlighted it in red.

Thank you again for your affirmation of our research.

Reviewer 2 Report

Comments and Suggestions for Authors

The authors have changed the text, but the structure of the manuscript remains unchanged. The authors did not even indicate the type of manuscript on line 1.

As a matter of common practice, there should be an introduction (including 1. literature review), results (including sections 2-6), a discussion of the results and a conclusion.

Therefore, I leave it up to the Editor to decide on this manuscript.

Author Response

Thank you again  for the precious suggestions, We highly value the feedback and suggestions you provided.

1.Reviewer Question 1: The authors have changed the text, but the structure of the manuscript remains unchanged.

Answer:  We merging the introduction and literature review section as requested, and added a discussion section.

2.Reviewer Question 2:The authors did not even indicate the type of manuscript on line 1.

Answer: We sincerely apologize for the oversight regarding the article type, and we greatly appreciate your commitment to academic rigor. 

3.Reviewer Question 3: As a matter of common practice, there should be an introduction (including 1. literature review), results (including sections 2-6), a discussion of the results and a conclusion.

Answer: We have reorganized the structure according to the requirements and added a discussion section. which describes as follows:

     From the experimental results, it is evident that the NN-like fuzzy controller outperforms the traditional fuzzy controller in terms of steady-state error, control stability, and response to parameter perturbations, particularly for the residual liquid outlet. However, the downside of the NN-like fuzzy controller is the significantly higher computational burden compared to the traditional fuzzy controller. Given the current hardware structure and the neural network’s layer and architecture, it can adequately support the computational load. Therefore, the NN-like fuzzy controller holds immense practical value and is expected to perform exceptionally well in practical applications.

4. All the revised parts have been marked in red.

Thank you very much for your valuable comments. Your academic rigor has been of great benefit to us for a lifetime.

Reviewer 3 Report

Comments and Suggestions for Authors

The revision manuscript "Application of Intelligent Control in Chromatography Separation Process" respond to requested suggestions and corrections" and can be considered for publish.

However, the graphic presentation must be improved. Some figures are dificult readable (ex. 1, 4 and 5)

Author Response

Thank you very much for the precious suggestions, according to your requirements, introduce some changes as follow:

1.Reviewer Question 1:The graphic presentation must be improved. Some figures are difficult readable (ex. 1, 4 and 5).

Answer:  According to your request, I have enhanced the pixel intensity of Fig. 1 and reprocessed images 4 and 5 to improve clarity and visualization.

2. All modified parts in the article are marked in red.

Thank you very much again for your valuable feedback.
